# Prevalence and Factors Associated with Sexual Violence against Children in a Brazilian State

**DOI:** 10.3390/ijerph19169838

**Published:** 2022-08-10

**Authors:** Márcia Regina de Oliveira Pedroso, Franciéle Marabotti Costa Leite

**Affiliations:** 1Center for Biological and Health Sciences, Federal University of Western Bahia, Barreiras 47810-047, Brazil; 2Pos-Graduation Program in Collective Health, Health Sciences Center, Federal University of Espírito Santo, Vitoria 29043-900, Brazil

**Keywords:** child abuse, violence, epidemiological monitoring, cross-sectional studies

## Abstract

Sexual violence is one of the forms of violence against children worldwide. Understanding its magnitude and its associated factors is essential to promote effective protection policies to childhood. The objective was to verify the prevalence and analyze the factors associated with sexual violence against children in a Brazilian state. This is a cross-sectional study analyzing data from reported cases of violence against children in the state of Espírito Santo, Brazil, between 2011 and 2018. The characteristics of the victim, perpetrator and aggression were studied, and the associations were analyzed using Poisson regression. The frequency of sexual violence was 41.8% and was more prevalent in girls, in the age groups 3 to 5 and 6 to 9 years old, in white ethnicity/color and in the urban area. The offenders were mainly men, known to the victim and occurred mainly in the residence. Sexual violence was the most reported violence among children in Espírito Santo, occurring within their circle of trust, demonstrating the need to provide support for families and to advance public policies to guarantee children’s rights.

## 1. Introduction

Worldwide, children are one of the most vulnerable groups to suffer violence due to their inherent fragilities and vulnerabilities, their physical, emotional and financial dependence and their low capacity for reaction and resistance [1,2]. One of the main forms of violence that victimizes childhood is the sexual type.

The Brazilian Ministry of Health (MH) considers sexual violence every act of sexual nature aimed at satisfaction, in which the offender is at a psychosocial level higher than that of the child [3]. Therefore, any behavior that affects children’s sexual development should be considered violence, such as rape, incest, sexual harassment, sexual exploitation, pornography, among others [3]. Furthermore, sexual violence can occur regardless physical contact and penetration, such as exposure to lewd acts and use of erotic language.

Due to their stage of growth and development, sexual acts are not understood by the child and, therefore, they cannot be consented, since the child is not aware of its implications [4,5]. Several studies indicate that the experience of sexual violence in childhood can lead to several consequences, such as emotional deficits, impulsive behaviors, hypersexualization, mental disorders, alcohol and drug abuse, among other situations that will affect children’s current and future life, and also the society as a whole [6,7,8]. It is highlighted that these situations of violence are the result of a sexist, patriarchal and racist culture present in Brazilian society [9].

According to data from the United Nations Children’s Fund (UNICEF), about 17 million adult women from 38 low and middle-income countries were victims of forced gender during their childhood [10]. Data from the World Health Organization (WHO) indicate that 20% of girls suffered sexual violence during childhood, which can reach 33% in some countries, against 7.6% of boys [11].

In Brazil, an analysis conducted from the Notifiable Diseases Information System (SINAN) between 2011 and 2017 showed that 58,037 reported cases of sexual violence against children, with an increase of 64.6% during the years. Most of the victims were girls, aged between 1 and 5 years old, black, without disabilities and/or disorders and who lived in the Southeast region [12]. Data from other localized studies indicate prevalence ranging from 16.4% [13] to 36.7% [14]. However, this magnitude can be wrong due to the difficulty of children in revealing abuse while still in childhood [15,16]. This is a very delicate subject that remains hidden and, therefore, unreported.

The notification is mandatory according to the Statute of the Child and Adolescent (ECA), and is another tool to accomplish child’s rights [17]. It is a trigger of the care network and an essential tool to put an end to the cycle of violence. In the health sector, Surveillance System for Violence and Accidents (VIVA) was implemented in 2006, also creating a specific form for the notification of violence cases in all age groups to be used throughout the national territory and mandatory completion by all professionals, establishing the significant role of this sector in the prevention and monitoring of victims [18]. This instrument has become an important source of data on the situation of violence in Brazilian population, serving as a basis to formulate policies and organize health services [19].

This study aims to identify the prevalence of sexual violence cases against children reported in the state of Espírito Santo, Brazil, and their association with characteristics of the victim, offender and offense.

## 2. Materials and Methods

This is a cross-sectional study in which all cases of sexual violence reported in the Notifiable Grievances Information System (SINAN) of the state of Espírito Santo were analyzed from 2011 to 2018. The period in question was chosen because violence became on the list of compulsory notification injuries in 2011. The data were provided by the State Department of Health (SESA).

Espírito Santo is located in the Southeastern Brazilian region, with an estimated population of 509,336 children in 2019, representing 14.5% of the 4,018,650 inhabitants. It is divided into 78 municipalities, with only nine of them presenting more than 100,000 inhabitants. It has a Human Development Index (HDI) of 0.740, considered high, and an average *per capita* income of R$1477.00, value close to the minimum wage (R$1212.00) [20].

All reported cases of sexual violence with children between 0 and 9 years old were selected to our analysis. It is highlighted that these are cases that sought health services (primary care units, hospitals, among others), spontaneously or by appointment. Then, the health professionals found or suspected the situation of sexual violence and filled out the notification form, as this is compulsory. These forms are then forwarded to the epidemiological surveillance sector of the municipalities, which register them in the SINAN system.

After the selection of these records, the qualification process of the database was caried out to correct possible errors and inconsistencies [18]. The registration field of the notification form number was evaluated for verification of duplications. In addition, each record was individually evaluated to see if other fields contained information that was filled in as ignore or that was blank in the field of interest—for example, details about the occurrence that were described in the observation field.

Sexual violence was analyzed as an outcome (no/yes). The characteristics of the victim, offender and offense were included as independent variables, which were categorized as follows: victim’s gender (male; female); age (0 to 2 years old; 3 to 5 years old; 6 to 9 years old); ethnicity/color (white; black or mixed race); presence of disabilities and/or disorders (no; yes); area of residence (urban/periurban; rural); offender’s age group (0 to 19 years old; 20 years old or more); gender (male; female; both); bond between offender and victim (father/stepfather/mother/stepmother/both parents; known—brothers, sisters, uncles, aunts, grandfather, grandmother, friends; unknown); suspected use of alcohol (no; yes); number of people involved (one; two or more); occurrence in residence (no; yes); period of occurrence (morning/afternoon; night/dawn); recurrent violence (no; yes) and referral to other network services (no; yes).

The descriptive analysis of the variables evaluated the relative and absolute frequencies and 95% confidence intervals. Pearson’s Chi-Square test was used in the bivariate analysis. For the multivariate analysis, a hierarchical model was used, with the characteristics of the victim as the first level and the characteristics of the offender and offense as the second level. The entry of the variables in the model respected the criterion of *p* < 0.20 in the bivariate analysis and for its maintenance the criterion of *p* < 0.05. In this model, Poisson Regression was used with an estimate of prevalence ratios. Analyses were performed in the Stata 14.1 program.

All ethical criteria defined by Resolution No. 499/2012 of the National Health Council were respected. This study was approved by Research Ethics Committee of the Universidade Federal do Espírito Santo under Opinion no. 2.819.597).

## 3. Results

### 3.1. Description

We analyzed 3127 reports of violence against children, being 1290 (P: 41.8%; CI 95%: 40.0–43.5) cases of sexual violence. Sexual violence was the most reported violence against girls, with 957 cases, corresponding to 54.6% (CI 95%: 52.3–56.9%) of cases. Among boys, it was the third most reported type of violence, with 333 cases, corresponding to 24.9% (CI 95%: 22.6–27.3%) (data not shown in table). Table 1 presents the common characteristics of cases of sexual violence.

### 3.2. Associated Factors

In the bivariate analysis (Table 2), sexual violence was related to victim’s gender, age group and ethnicity/color, offender’s area of residence, age group and gender, bond with the victim, suspected alcohol use, number of involved, occurrence in the residence and referral to other services (*p* < 0.05).

In the adjusted analysis, sexual violence remained associated with victim’s the gender, age and ethnicity/color, the area of residence, offender’s gender, the bond with the victim and the occurrence in the residence (Table 3).

## 4. Discussion

Sexual violence was present in more than 40% of the notifications recorded in Espírito Santo from 2011 to 2018, being the violence that most victimized children. The abuse was more frequent in girls aged three years old, white, and residents of the urban area. The offenders are mainly male, from the child’s social circle and the main place of occurrence is residence.

This study found a prevalence for sexual violence reports higher than all studies that also analyzed SINAN data in Brazil. It was the most reported violence in the period studied, which was also found by Oliveira et al. (2020) analyzing data from 2009 to 2016 in municipality of Manaus [21]. Malta et al. (2017) analyzing data from hospital emergency services across the country from the VIVA Survey found that sexual violence was the third most frequent harm, behind neglect and physical violence [22]. Rates et al. (2015) when analyzed nationwide SINAN data found a prevalence of 37% for sexual violence, this result is closer to that we found in our study [23]. The authors believe that these differences are due to a higher qualification of the Espírito Santo database and also because this state had the highest levels of violence in Brazil during the period studied [24].

The main victims of sexual violence in childhood are girls, as demonstrated in this study and in the scientific literature [21,22,23,25]. This situation reflects the gender as a social structure, which culminates in a historical inequality between men and women, making women more vulnerable to all types of violence, especially sexual violence, which is repeated in several countries and cultures, regardless their age [26,27]. Society sees women as fragile and submissive, being considered men’s satisfaction objects. We must understand these different power relations and transform the social norms that stereotype women and children [28,29]. We also inferred that the proportion of boys is underreported due to the prejudices and stigmas they suffer after violence due to their masculinity and gender identity. This bias, therefore, silences their suffering [30].

The data of this study, and those indicated in literature, indicate that children ≥3 years old as the most vulnerable ages to suffer sexual violence, especially from six to nine years old [13,31,32]. The literature point that the younger the child, the more difficult to reveal the violence suffered, especially considering their lower language development [15,33]. Arrendondo et al. (2016), observing data from 886 cases in Chile, found that the percentages of disclosure of sexual violence increases with the increase of the child’s age, becoming higher than those of detection (when another person reports the abuse), which is the most common situation involving underage victims [34].

The literature differs on which ethnicity is most affected by sexual violence in childhood, which also depends on where the research was conducted. In this study, white children had a 10% higher prevalence when compared with black and mixed-race children. We believed that this situation is due to the ethnic composition of the population of Espírito Santo, which, according to the 2010 Census, is composed mostly of white people [20]. Platt et al. (2018), analyzing data from SINAN from the municipality of Florianópolis, state of Santa Catarina, also found a higher prevalence of white children [35]. Santos et al. (2018), studying data on sexual violence against children and adolescents that occurred at school and that were registered in the SINAN throughout Brazil, also found a higher prevalence of white children [31].

This study shows that the prevalence of sexual violence in childhood in the urban area was higher than in the rural area. This prevalence was also found in Florianópolis [35], and in the north of the state of Minas Gerais [13]. These results can reflect the structural violence that is strongly present in urban areas and the difficulty of access to health services in the rural area, besides cultural issues that hinder the revelation of the violence [36,37].

When we talk about sexual violence, the main offenders are men, as indicated in this and other studies [25,35,38], demonstrating once again the gender inequalities and generational hierarchy existing in Brazilian society. The offenders, in turn, claim to be “seduced” by the child, in an attempt to justify an unjustifiable act, which another demonstration of power exercised by men [39].

Regarding the higher prevalence of the category “both gender,” we believe that this is more due to the connivance of females regarding the situation of sexual violence suffered by the child than by the presence of offenders of both gender, since no studies indicating this situation were found in the literature. What is seen is that women, especially those who represent the maternal role, become conniving with the situation experienced by the child, not only for fear of losing their partner, but also as an attempt to preserve the family [2,6], since one of the main measures adopted after the complaint is the removal of the offenders from the child’s household. Moreover, the fact that the child is a victim of violence can also generate feelings of failure in the mother, since maternity is related to the role of protection and care [29,40]. This omission also represents violence against children since neglect hampers the denounce and contributes to keep children in the violent cycle.

Regarding the main offenders, literature points to the paternal figure (fathers and stepfathers) as the main ones [21,22]. However, this study shows that the offenders were people known to the victim, without parental bond, such as uncles, grandparents, cousins and brothers. Other studies have also pointed to those known as the main offenders [8,13,25]. These people are also important in the child’s circle and, in general, have unrestricted access to the victim’s residence, often acting as caregivers, possessing the affection and trust of the infant and family. Miranda et al. (2014) also indicate that often the offender can be declared as “known” for fear of denouncing the real culprit [41].

The offenders has advantage over the child since they are inside the child’s trust circle [5]. This one then takes advantage of the victim’s innocence, initiating violent acts in less invasive manners. By playing, the offender conducts child to think that these play is a demonstration of affection [2,16,33]. Then, when child gives total confidence to the offender, the violence increases in frequency and severity. At this moment, the child begins to realize that these acts are not correct, and the offender starts threatening, resulting in the child feeling guilty about what is happening [32,42]. This demonstrates once again the abuse of adult power and the generational inequality in the sexual violence against children, allowing the offender to take advantage of the child’s vulnerability through extortion, intimidation and silence [6,15,33]. Lira et al. (2017) indicate that fear is one of the main reasons why victims do not report the violence that has been happening, mainly because they fear that the offender will do something more serious with them or their families [6].

Studies conducted throughout Brazil indicate residency as the main place of sexual violence against children, as we also showed in this study [21,32,35]. This is a paradox, since the family environment should be the place where the child feels safe, loved and protected and not vulnerable to any kind of rights violation [2]. The occurrence of violence in the household also contributes to cover such violence, due to the pacts of silence between residents and the culture of society that privilege the privacy of family environment, even if situations of rights violation [43]. Even when child tries to expose the violence suffered, he/she is often not heard or adults consider it to be a lie [5,33,44].

Vieira et al. (2015) highlight that communicating sexual violence is fundamental for its confrontation, for protection and assistance to families and interruption of the cycle [45]. Thus, the health sector has a fundamental role because it receives victims to deal with physical and psychological damage resulting from this violence [3]. Moreover, this sector, by Primary Care, also has direct contact with families, and can be an identifier of triggering violence situations or children who are already being victims [3,46]. The notification generated by health units acts as a trigger of the care network, which should act together to protect and assist infants who are victims of sexual violence. Professionals must be trained to identify violent situations and to fully act in the support of the families, as well as give emotional and psychological support necessary to deal with situations of violence, especially against children [3]. It is also essential that the professional has a qualified listening posture, so as not to generate more suffering and trauma for the victim.

As limitations of our study, we cite those related to the proper filling of notification forms and underreporting of cases, as the reported cases are those where victims sought health services. To minimize the effects of the first, we performed the database qualification process to reduce possible errors and inconsistencies. In this sense, health professionals must also be trained to correctly fill out the information. Concerning the underreporting, it reflects the taboos and myths involving this theme, a result of the sexist and patriarchal culture of Brazilian society. Therefore, the process of discussion about sexual violence and children’s rights must be broadened, so that more victims and families can reveal the occurrence and so that professionals from all areas are able to correctly identify cases.

## 5. Conclusions

Sexual violence was the most reported violence among children from the state of Espírito Santo, being more prevalent in those older than three years old, white and urban. This occurred mainly within the child’s circle of trust, at home and perpetrated by people of child’s coexistence.

In this interim, it is important to track and notify this outrage to break down the sexual violence cycle. We must also consider the effects of sexual violence on child’s life and health, and the need for further studies to better understand this phenomenon. The occurrence of sexual violence raise awareness of the whole society, encouraging for the protection of children and for the notification of cases, and should be considered a collective problem. Thus, it will be possible to design and implement policies that are really effective and intersectoral; only then childhood will have its rights restored, without any form of violence.

## Figures and Tables

**Table 1 ijerph-19-09838-t001:** Characterization of reported cases of sexual violence against child according to data from the victim, the offender and the offense. Espírito Santo, 2011 to 2018.

Variables	n	%	CI 95%
**Victim’s gender**			
Male	333	25.8	23.5–28.3
Female	957	74.2	71.7–76.5
**Victim’s age group**			
0 to 2 years old	252	19.7	17.6–22.0
3 to 5 years old	456	35.7	33.1–38.3
6 to 9 years old	571	44.6	41.9–47.4
**Ethnicity/Color**			
White	335	30.4	27.7–33.2
Black/Mixed Race	768	69.6	66.8–72.3
**Deficiencies/Disorders**			
No	1187	96.3	95.1–97.2
Yes	46	3.7	2.8–5.0
**Place of residence**			
Urban/Periurban	1178	93.3	91.8–94.5
Rural	85	6.7	5.5–8.3
**Offender’s age group**			
0–19 years old	221	36.6	32.8–40.5
20 years or more	383	63.4	59.5–67.2
**Offender’s gender**			
Male	1019	91.6	89.8–93.1
Female	49	4.4	3.3–5.8
Both	45	4.0	3.0–5.4
**Bond with the victim**			
Father/Stepfather/Mother/Stepmother/Both Parents	385	35.1	32.3–38.0
known	675	61.5	58.6–64.4
Unknown	37	3.4	2.5–4.6
**Suspected use of alcohol**			
No	474	83.9	80.6–86.7
Yes	91	16.1	13.3–19.4
**Number of involved**			
One	937	86.4	84.2–88.3
Two or more	148	13.6	11.7–15.8
**Occurred at the residence**			
No	130	12.1	10.2–14.2
Yes	948	87.9	85.9–89.8
**Period of Occurrence**			
Morning/Afternoon	367	65.5	61.5–69.4
Night/Dawn	193	34.5	30.6–38.5
**Repeated violence**			
No	355	43.6	40.2–47.0
Yes	460	56.4	53.0–59.8
**Referral**			
No	72	5.6	4.5–7.0
Yes	1206	94.4	93.0–95.5

**Table 2 ijerph-19-09838-t002:** Distribution of sexual violence perpetrated against child according to the characteristics of the victim, the offender and the offense. Espírito Santo, 2011 to 2018.

Variables	n	%	CI 95%	*p*-Value
**Victim’s gender**				
Male	333	24.9	22.6–27.3	<0.001
Female	957	54.6	52.3–56.9	
**Victim’s age group**				
0 to 2 years old	252	23.7	21.2–26.3	<0.001
3 to 5 years old	456	51.9	48.6–55.2	
6 to 9 years old	571	51.1	48.2–54.1	
**Ethnicity/Color**				
White	335	45.9	42.3–49.5	0.030
Black/Mixed Race	768	41.2	39.0–43.5	
**Deficiencies/Disorders**				
No	1187	41.7	39.9–43.5	0.974
Yes	46	41.8	32.9–51.3	
**Place of residence**				
Urban	1178	42.9	41.1–44.8	0.002
Rural	85	32.8	27.4–38.8	
**Offender’s age group**				
0–19 years old	221	68.0	62.7–72.9	<0.001
20 years old or more	383	37.3	34.4–40.3	
**Offender’s gender**				
Male	1019	68.9	66.4–71.2	<0.001
Female	49	6.4	4.8–8.3	
Both	45	8.5	6.4–11.2	
**Bond with the victim**				
Father/Stepfather/Mother/Stepmother/Both Parents	385	22.3	20.4–24.3	<0.001
known	675	70.8	67.9–73.6	
Unknown	37	37.4	28.4–47.3	
**Suspected use of alcohol**				
No	474	43,1	40.2–46.0	<0.001
Yes	91	27.6	23.0–32.7	
**Number of involved**				
One	937	46.3	44.2–48.5	<0.001
Two or more	148	19.9	17.2–22.9	
**Occurred at the residence**				
No	130	25.4	21.9–29.4	<0.001
Yes	948	43.3	41.3–45.4	
**Period of Occurrence**				
Morning/Afternoon	367	39.5	36.4–42.7	0.633
Night/Dawn	193	38.2	34.1–42.5	
**Repeated violence**				
No	355	44.5	41.1–48.0	0.522
Yes	460	46.0	42.9–49.1	
**Referral**				
No	72	20.0	16.2–24.5	<0.001
Yes	1206	44.7	42.8–46.6	

**Table 3 ijerph-19-09838-t003:** Crude and adjusted analysis of the effects of the characteristics of the victim, the offender and the offense with the sexual violence perpetrated against children. Espírito Santo, 2011 to 2018.

Variables	Crude Analysis			Adjusted Analysis		
	PR	CI 95%	*p*-Value	PR	CI 95%	*p*-Value
**Victim’s gender**						
Male	1.0		<0.001	1.0		<0.001
Female	2.20	1.98–2.43		2.10	1.88–2.34	
**Victim’s age group**						
0 to 2 years old	1.0		<0.001	1.0		<0.001
3 to 5 years old	2.19	1.93–2.48		2.12	1.86–2.42	
6 to 9 years old	2.16	1.91–2.44		2.15	1.89–2.45	
**Ethnicity/Color**						
White	1.11	1.01–1.23	0.027	1.11	1.02–1.22	0.018
Black/Mixed Race	1.0			1.0		
**Place of residence**						
Urban	1.31	1.09–1.57	0.003	1.37	1.15–1.64	<0.001
Rural	1.0			1.0		
**Offender’s age group**						
0–19 years old	1.82	1.63–2.03	<0.001	1.05	0.92–1.20	0.450
20 years old or more	1.0			1.0		
**Offender’s gender**						
Male	10.83	8.24–14.24	<0.001	7.90	5.72–10.90	<0.001
Female	1.0			1.0		
Both	1.34	0.91–1.98		1.67	1.09–2.55	
**Bond with the victim**						
Father/Stepfather/Mother/Stepmother/Both Parents	1.0		<0.001	1.0		<0.001
Known	3.18	2.88–3.50		1.59	1.45–1.74	
Unknown	1.68	1.28–2.20		1.23	0.93–1.64	
**Suspected use of alcohol**						
No	1.56	1.29–1.88	<0.001	1.18	0.99–1.38	0.054
Yes	1.0			1.0		
**Number of involved**						
One	2.33	2.00–2.71	<0.001	0.92	0.76–1.11	0.369
Two or more	1.0			1.0		
**Occurred at the residence**						
No	1.0		<0.001	1.0		<0.001
Yes	1.7	1.46–1.99		1.47	1.27–1.70	

## Data Availability

Not applicable.

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
