# Peer review of "Prevalence and Factors Associated with Sexual Violence against Children in a Brazilian State"

_ijerph, 2022, doi:10.3390/ijerph19169838_

Round 1

Reviewer 1 Report

Thank you very much for the opportunity to read this paper. It is critical to expand research on the magnitude and associated factors of sexual violence against children in low- and middle-income countries. This paper addresses an extremely importance area of research. I found it very informative. I hope the comments below would be helpful.

Overall

1.     I think it is necessary to provide readers with more contextual information, for example, by explaining more about local cultural aspects that may contribute to sexual VAC perpetration and reporting in Brazil. It will enable readers to understand why and how violence should be understood within contexts, as well as how the situation may differ between high-income countries/regions and LMICs. It will also allow them to decide how generalisable the findings are to their own contexts.

2.     In the paper, “sexual violence” and “physical violence” are used interchangeably, for example in the abstract line 14, research objective line 63, and Table 3. Could you please correct and/or clarify?

3.     I would suggest checking editorial, syntactic, and grammar issues.

Abstract

Line 15: Could you please clarify the meaning of ‘author’?

Introduction

1.     Line 31-32: I understand the importance of conceptualising sexual violence within the local culture and context and the reason why you use a Brazilian definition of sexual violence. However, I am concerned that the definition you provide here—“any behavior that affects sexual develop”— may be too broad and not capture the feature of sexual violence. I wondered if it would be helpful to look at the international guidelines, e.g. UNICEF Hidden in Plain Sight, and discuss/justify your definition.   

2.     Line 33-34: Would it be helpful to give examples of non-contact sexual violence for readers who are not familiar with this concept?

3.     Line 47: Was it meant to be “58,037”?

4.     Line 63: Could you please clarify the research aim?

Methods

1.     Line 76: what does 75 average per capita income of R$1,477.00 mean in a Brazilian context—low, middle, or high income?

2.     Line 77: could you please explain the rationale of choosing this age range?

3.     Line 79: could you please elaborate how you identified and corrected possible errors?

4.     Line 85-86: Since the way you operationalized offender’s age (=<19 and >=20 indicated that there might be offenders who were children, I was wondering why relationship between offender and victim did not include e.g. siblings or peers? Were “known” and “Father/Stepfather/Mother/Stepmother/Both Parents” mutually exclusive?

5.     Line 87: could you please explain the meaning of “number of involved”?

Results

1.     Associate factors: I am not sure if the results of Pearson’s Chi-Square tests could be interpreted as evidence of relationships between outcome and covariates.

Discussion

1.     Were the studies you compared your findings to also conducted in Brazil?

2.     Line 139-140: In the results section, you mentioned that sexual violence was the most prevalent form of violence only for girls.

3.     It would be helpful to explain why the level of sexual violence you found is higher than the estimates in any other existing studies in Brazil.

4.     In terms of the association between child age and prevalence of violence, does the prevalence or occurrence genuinely increase with child age, or would it just be the number of cases that are reported that increases with child age?

5.     Could you please clarify whether your findings concerning ethnicity were consistent or inconsistent (both could make sense) with Platt 2018 and Santos 2018? I am not sure if Platt 2018 and Santos 2018 found higher prevalence rates among White children, or if they involved more While children but the prevalence rates experienced by the White children were not necessarily higher than other ethnic groups.

Author Response

The answers are in the attached file.

Reviewer 2 Report

Thank you for letting me review this manuscript. It analyzes a relevant issue, such as the child sexual abuse with official data. The authors include current literature and report interesting data about the CSA. However, I have some considerations that I hope may contribute to improve the quality of the manuscript.

Overall, the main issue to improve is providing more information about the methodology data collection. It doesn't remain clear along the manuscript how it was assessed the sexual abuse, who exactly inform o report the presence of this kind of abuse. Concretely, I comment specific aspects below:

- Title: It seems to be a very long title, with too many words. I recommend considering to remove the date for example. 

- Abstract (but also objectives and more sections): I suggest revise the main objectives and some aspects about "physical violence". I guess you are referring the data were extracted from the physical violence reports in children, but in the title it is named "sexual violence against children", in the abstract the objective is "to verify the prevalence and analyze the factors associated with violence against children in a Brazilian state", but the the results talk about sexual violence; and then in the objectives (page 2, line 63) it is stated that "This study aims to identify the prevalence of physical violence cases against children". This seems to be inconsistent, so it is stated different types of violence when referring the objectives of the study. I recommend to always be consistent and referring the same words and issue along the manuscript.

- Materials and Methods: here I found the most important need for improvement. Information about the instrument and methodology is lacked. How exactly has the presence/absence of sexual violence been assessed? It was a professional criterion? Victims disclosure? Familiars disclosure? Which professionals assessed this issue? 

Could you explain more about the instrument? How it is reported the violence? By whom? What about the context? 

This is a very important information, because it could be influencing importantly in the limitations and conclusions. If we are only analyzing the cases reported by health professionals for example in hospital context, we are focusing in the more serious cases, and most cases are missing so we may not talk about "prevalence" in general...This should be included in the limitations, reflecting on sample bias and unrepresentative results of the general population.

- Results: Tables 1 and 2 include very similar data. I am not sure if it would be possible to fuse both tables including the most relevant information. For example, in table 2 the chi square data are not reported. The variables and the N for every variable condition is the same in both tables. I think data are presented in a redundant way. 

Also, in page 5 there is a lot of information repeated in the table and in the paragraph. Tables' data should not be duplicated in the text.

- Discussion: it is advisable to begin this section remembering the main aim of the study. In line 141 it is said that "The abuse was more frequent in girls aged three years old..." but in data the age is 3-5 years old. 

In the second paragraph it is said that this study finds "a prevalence for sexual violence reports higher than all studies that also analyzed SINAN data", what explanation consider the authors about this fact? what do you think are the reasons of this higher prevalence?

Along the discussion section the authors include many information about the disclosure difficulties, but in this manuscript this variable is not included or analyzed. I recommend to take care when talking about the prevalence too...this is the prevalence with a very important bias: these are data collected by formal or official data and it is known that the vast majority of the CSA cases are never report to the authorities. This could be include as a limitation of the study for talking about prevalence. Indeed, this could be the prevalence of the most severe cases.

It is a very interesting manuscript that contributes to the scientific knowledge about CSA, specifically those reported by professionals. The considerations for future prevention and professional trainings are very valuable.

Author Response

The answers are in the attached file.

Round 2

Reviewer 2 Report

I would like to thank the authors for the effort to improve the manuscript. There are still some considerations to take into account:

- in the Materials and Methods section I still find a lack of information about the instrument. It continues without including a more exhaustive explanation of how the data was collected. Understanding the data collection is essential to analyze the results. If they could include more information about the form that professionals fill out, it would greatly improve the manuscript.

- About the information repeated in tables and text, I insist in the APA recommendation: "The Publication Manual states that effective tables and figures supplement or augment the text rather than duplicate it (see pp. 130 and 152)."

- In page 6, the text included in the revision is not in English language. Furthermore, this is the only explanation added to the suggestion to develop the considerations on the highest prevalence found, that I find insufficient.

Author Response

Dear Reviewer, 

We made the changes suggested in the article “Prevalence and factors associated with sexual violence against children in a Brazilian state, from 2011 to 2018”. Below are responses to each item.

Sincerely,
The authors
Reviewer 2
- in the Materials and Methods section I still find a lack of information about the instrument. It continues without including a more exhaustive explanation of how the data was collected. Understanding the data collection is essential to analyze the results. If they could include more information about the form that professionals fill out, it would greatly improve the manuscript.
Resposta: The authors are grateful and the text has been changed.
- About the information repeated in tables and text, I insist in the APA recommendation: "The Publication Manual states that effective tables and figures supplement or augment the text rather than duplicate it (see pp. 130 and 152)."
Resposta: The authors are grateful and the text has been changed.

- In page 6, the text included in the revision is not in English language. Furthermore, this is the only explanation added to the suggestion to develop the considerations on the highest prevalence found, that I find insufficient.
Resposta: The authors are grateful and the text has been changed.